# Unraveling the Role of Epithelial Cells in the Development of Chronic Rhinosinusitis

**DOI:** 10.3390/ijms241814229

**Published:** 2023-09-18

**Authors:** Jong-Gyun Ha, Hyung-Ju Cho

**Affiliations:** 1Department of Otorhinolaryngology—Head and Neck Surgery, Chung-Ang University Gwangmyeong Hospital, Gwangmyeong 14353, Republic of Korea; jonggyunha@cau.ac.kr; 2Department of Otorhinolaryngology, Yonsei University College of Medicine, Seoul 03722, Republic of Korea; 3The Airway Mucus Institute, Yonsei University College of Medicine, Seoul 03722, Republic of Korea

**Keywords:** epithelial cell, nasal polyp, rhinosinusitis, nasal, immune, interaction

## Abstract

The pathophysiology of CRS is multifactorial and complex yet needs to be completed. Recent evidence emphasizes the crucial part played by epithelial cells in the development of CRS. The epithelial cells act as physical barriers and play crucial roles in host defense, including initiating and shaping innate and adaptive immune responses. This review aims to present a comprehensive understanding of the significance of nasal epithelial cells in CRS. New research suggests that epithelial dysfunction plays a role in developing CRS through multiple mechanisms. This refers to issues with a weakened barrier function, disrupted mucociliary clearance, and irregular immune responses. When the epithelial barrier is compromised, it can lead to the passage of pathogens and allergens, triggering inflammation in the body. Furthermore, impaired mucociliary clearance can accumulate pathogens and secretions of inflammatory mediators, promoting chronic inflammation. Epithelial cells can release cytokines and chemokines, which attract and activate immune cells. This can result in an imbalanced immune response that continues to cause inflammation. The interaction between nasal epithelial cells and various immune cells leads to the production of cytokines and chemokines, which can either increase or decrease inflammation. By comprehending the role of epithelial cells in CRS, we can enhance our understanding of the disease’s pathogenesis and explore new therapeutics.

## 1. Introduction

Chronic rhinosinusitis (CRS) is a prevalent and debilitating inflammatory condition of the nasal and paranasal sinus mucosa, affecting millions of people worldwide. CRS is characterized by persistent inflammation, leading to sinonasal symptoms that last for at least 12 weeks. Based on clinical presentation, CRS can be further classified into two main phenotypes: CRS with nasal polyps (CRSwNP) and CRS without nasal polyps (CRSsNP) [1]. Research on the pathophysiology of CRS suggests that phenotypes alone are insufficient for the appropriate classification of CRS.

Consequently, there has been a shift toward studying endotypes, which are defined by distinct molecular and immunologic profiles [2]. Eosinophilic CRS (ECRS) and non-eosinophilic CRS (non-ECRS) are the two essential endotypes of CRSwNP. ECRS is characterized by the presence of eosinophils, which play a key role in the pathogenesis of the disease. In ECRS, the nasal epithelium undergoes remodeling accompanied by goblet cell hyperplasia, submucosal gland hyperplasia and hypertrophy, and basement membrane thickening, as seen in the airway remodeling effects of asthma [3]. In addition, there is an increased expression of T-helper 2 (Th2) cytokines and chemokines, which contribute to eosinophil recruitment and activation. On the other hand, non-ECRS is characterized by a lower number of eosinophils and a different inflammatory profile, with an increased expression of Th1 and Th17 cytokines and chemokines. These endotypes help elucidate the complex pathophysiology of CRS and may guide personalized treatment strategies.

Despite extensive research, the pathophysiology of CRS remains incompletely understood. Recent evidence highlights the critical role of epithelial cells in the pathophysiology of CRS. As our understanding of the pathophysiology of CRS has evolved, the sinonasal epithelium has been transformed from a simple passive barrier into an active immunologic organ that participates in both innate and adaptive immune responses [4].

The sinonasal epithelium is a complex tissue that serves as a physical and immunological barrier against environmental insults and pathogens and is also involved in modulating both innate and adaptive immune responses [5]. Dysregulation of the nasal epithelium, including impairment of the epithelial barriers and altered interactions with immune cells and the extracellular matrix (ECM), has been implicated in the pathogenesis of CRS. This review aims to provide an overview of the current understanding of the role of nasal epithelial cells in CRS, with a particular focus on their histological features, barrier functions, and crosstalk with other immune cells and the ECM.

## 2. Overview of Sinonasal Epithelial Cell Subtypes

The sinonasal epithelium is one of the first barriers exposed to external environmental stimuli such as pollutants, cigarette smoke, allergens, and microbes. It is a pseudostratified columnar epithelium composed of various cell types, including ciliated cells, goblet cells, basal cells, and nonciliated columnar cells [6]. This specialized epithelium plays a critical role in maintaining the health of the respiratory tract by acting as a physical barrier and participating in mucociliary clearance.

Ciliated cells, the dominant occupants of this epithelium, flaunt hundreds of motile cilia, along with a greater number of immotile microvilli, on their apical surface. Microvilli increase the surface area of the cell, facilitating exchange with the external environment and preventing dehydration. By orchestrating the movement of mucus and entrapped particles, ciliated cells ensure that these are directed toward the nasopharynx for elimination.

Goblet cells secrete mucus, which serves as a barrier to trap inhaled particles and pathogens. Each cell contains abundant secretory granules that store mucin, which is released via a glandular duct on the cell’s apical surface [7]. Goblet cells, while not present in squamous or olfactory epithelium, are distributed irregularly throughout the sinonasal cavity [8,9].

Lastly, basal cells, anchored closest to the basal membrane and away from the epithelial surface, play the role of progenitors. Basal cells exhibit specific morphologic features, such as desmosomes, that provide strong adhesion between adjacent cells. Basal cells act as progenitor cells and are responsible for the regeneration of the epithelium [10].

While the roles of ciliated cells, goblet cells, and basal cells in the sinonasal epithelium are well established, recent advances in single-cell RNA sequencing (scRNA-seq) techniques have provided new insights about the cellular composition and diversity of the sinonasal epithelium, revealing various types of epithelial cells in the human respiratory epithelium (Figure 1). These uncharted cell subpopulations include mature multiciliated cells (expressing FOXJ1 and acetyl-α-tubulin), deuterosomal cells (FOXJ1, DEUP1, PLK4, CCNO, and CEP78), and secretory cells, like club cells (KRT7, SCGB1A1) and goblet cells (MUC5AC, KRT7, SCGB1A1). Additionally, other identified cells include ionocytes (CFTR, ASCL3, and FOXI1), suprabasal cells (SERPINB4 and KRT5), and cycling (MKI67+, KRT15, KRT5) and non-cycling (MKI67−, KRT5, TP63) basal cells [11,12,13].

In summary, multiciliated cells possess multiple cilia on their surface, which play a critical role in the movement of mucus and trapped particles out of the sinuses. Their primary function revolves around maintaining effective mucociliary clearance. Deuterosomal cells have only recently been reported in relation to airway multiciliogenesis. Deuterostomes are involved in the production of multiple cilia in multiciliated cells. Club cells are secretory cells also known as Clara cells. Besides their detoxifying actions against airborne threats, they maintain the epithelial barrier and exhibit impressive regenerative abilities. Ionocytes manage the secretion and absorption of ions to ensure that mucus has the right viscosity and that the epithelium stays hydrated. This is a relatively newly discovered cell type that is sparse but appears to play a crucial role in cystic fibrosis. Suprabasal cells, seated just above the basal layer, may contribute to barrier integrity and possibly partake in immune functions.

In addition to providing a better understanding of the transcriptional profiles of individual cell types, scRNA-seq has revealed distinct subsets of ciliated cells, goblet cells, and basal cells, each with unique gene expression patterns and functional roles [14,15].

## 3. Histological Changes and Tissue Remodeling in CRS

### 3.1. Histological Characteristics of CRS

CRS is characterized by various histological changes in the sinonasal epithelium, including tissue remodeling and the epithelial-to-mesenchymal transition (EMT). These alterations contribute to the pathogenesis of the disease and can differ between CRSsNP and CRSwNP.

The histopathological hallmark of CRSsNP is chronic inflammation, usually characterized by a preponderance of neutrophils and mononuclear cells (T-lymphocytes and monocytes) [16]. In CRSsNP, the nasal epithelium undergoes basement membrane thickening and exhibits excessive numbers of goblet cells, submucosal gland hyperplasia and hypertrophy, and fibrosis. Goblet cell hyperplasia leads to excessive mucus production and impaired mucociliary clearance [17]. This can result in mucus accumulation in the sinonasal cavity and persistent inflammation. Basement membrane thickening can affect epithelial cell regeneration and further compromise the epithelial barrier [18]. Furthermore, an infiltration of inflammatory cells, such as lymphocytes, eosinophils, and neutrophils, can be observed in the subepithelial layer, contributing to localized inflammation [19]. Tissue remodeling, including fibrosis, angiogenesis, and the excessive deposition of ECM proteins including collagen, is also commonly observed in CRSsNP, contributing to structural and functional changes in the sinonasal epithelium [18,20].

In contrast, CRSwNP is characterized by a distinct histopathological pattern. The polypoid tissue is marked by extensive edematous stroma, the formation of pseudocysts, intense inflammatory cell infiltration including eosinophils (particularly in Western countries), and the presence of Th2 cells [21,22,23]. Epithelial disruption and the loss of ciliated cells are commonly observed, which can impair mucociliary clearance and contribute to persistent inflammation [17]. Stromal tissue edema and basement membrane thickening are more frequently observed in CRSwNP than in CRSsNP, but the presence of stromal tissue fibrosis is not statistically different between these two CRS phenotypes [24].

### 3.2. Tissue Remodeling

Tissue remodeling in CRSwNP is characterized by the formation of pseudocysts filled with plasma protein, enhanced edema, and reduced ECM deposition compared to CRSsNP [18], resulting in less fibrotic and more edematous polyps, which contributes to the formation of nasal polyps. In CRSwNP, there is a change in the composition of various epithelial cell types, such as a decrease in the number of ciliated cells and an increase in the numbers of tuft cells, goblet cells, and mast cells. Such epithelial remodeling is driven by the activation of interleukin (IL)-13 and prostaglandin E2. In contrast, IL-17 is more prominent in CRSsNP, without accompanying changes in cell-type composition [25]. The innate immune system, which is composed of phagocytic cells like macrophages, dendritic cells (DCs), and polymorphonuclear granulocytes as well as mediators like cytokines and chemokines, in addition to the activation of the complement cascade, could contribute to the remodeling process [26,27,28].

### 3.3. EMT in CRS

Resulting in the transition of cells from epithelial to mesenchymal states, EMT is integral to wound healing, fibrosis, and tissue remodeling [29]. This is evident in CRSwNP, where EMT has been shown to be associated with epithelial barrier dysfunction, polyp formation, and tissue remodeling [20]. During EMT, epithelial cells undergo significant changes: the loss of apical-basal cell polarity and junctions with adjacent cells, gain of epithelial cell migratory capacity, and reorganization of the cytoskeleton such as the acquisition of vimentin filaments are hallmarks of the EMT process [30]. Simultaneously, affected epithelial cells acquire mesenchymal properties, including less rigid cell adhesion and a depolarized arrangement of the cytoskeleton [31]. Junctional proteins like E-cadherin, claudin, occludin, and zonula occludens 1 (ZO-1) are present at reduced levels in CRS, while matrix proteins and proteases such as periostin, laminin, vimentin, matrix metalloproteinases (MMPs) 2 and 9, fibronectin, and tenascin-C have increased expression levels in CRS [21].

Transforming growth factor-beta (TGF-β) has emerged as a cardinal modulator in the EMT paradigm. Epithelial markers, particularly E-cadherin, undergo suppression catalyzed by transcriptional repressors, such as SNAIL and TWIST, while mesenchymal markers like vimentin are concurrently upregulated [32]. TGF-β regulates the transcription of target genes promoting EMT by activating Smad2/3, leading to the formation of a complex with Smad4 [33]. Beyond the confines of canonical SMAD signaling, TGF-β delves into non-canonical avenues, including the PI3K/AKT and MAPK pathways [34] Additionally, TGF-β exhibits roles that transcend transcription regulation, extending to an architectural remodeling of the ECM via elevated MMP-9 and suppressed tissue inhibitors of metalloproteinases (TIMPs) [35].

Although TGF-β occupies a central position, EMT in CRS is also sculpted by molecular pathways that function independent of TGF-β. Noteworthy among these are the pathways modulated by molecular entities, such as Notch, Wnt, and HIF-1α [29,36,37]. Notably, CRSwNP predominantly involves the TGF-β/Smad signaling pathway, coupled with the eosinophil infiltration of the periosteum, suggesting their role in bone remodeling processes in CRSwNP. In stark contrast, CRSsNP operates via TGF-β/Smad-independent pathways [38].

Differences in the expression of epithelial and mesenchymal markers are more pronounced in CRSwNP than in CRSsNP. EMT is a prominent characteristic of CRSwNP, whereas its significance in CRSsNP remains elusive [37]. MicroRNAs have emerged as influential regulators in this context. For instance, alterations in miR-34 and miR-449 families have been found to be correlated with impaired epithelial cilium function in CRSwNP [39]. The expression levels of miR-21 and TGF-β1 mRNAs appear to be higher in CRSwNP than in CRSsNP, and researchers have reported that treatment with mir-21 inhibitor suppresses TGF-β1-induced EMT in primary human nasal epithelial cells (pHNEC) [40]. Moreover, miR-761 critically regulates the balance between N-cadherin and E-cadherin, which in turn impacts nasal mucosal remodeling via the LCN2/Twist1 pathway in mice [41]. Sirtuin 1 (SIRT1) could be another important factor in determining these distinct phenotypes. Interestingly, while the inflammatory environment of CRSsNP stimulates SIRT1, which is known to suppress HIF-1α-driven EMT, the nasal polyp mucosal conditions seem to lead to the loss of SIRT1 activation. A decrease in SIRT1 levels might amplify EMT, potentially driving sinonasal polyp formation [42]. Meanwhile, research suggests that the downregulation of miR-155-5p elicits the inhibition of EMT in pHNEC by targeting SIRT1 [43]. MicroRNAs could serve as regulators in these pathways, potentially guiding the phenotype and progression of CRS, making them not just markers but potential therapeutic targets.

In conclusion, EMT contributes to the disruption of the sinonasal epithelial cell (SEC) barrier by downregulating the expression of junctional proteins and upregulating mesenchymal markers, compromising the integrity of the barrier and causing patients with CRS to be more susceptible to inflammation and infection. Further research on the molecular mechanisms underlying EMT in CRS and the development of targeted therapies to prevent or reverse EMT may help improve the management of this chronic inflammatory condition (Figure 2).

## 4. Sinonasal Barriers in CRS

### 4.1. *Mucociliary Clearance*

Airway surface liquid (ASL) is a fluid of the epithelial lining that is composed of a superficial mucous layer and an underlying periciliary fluid layer. Most inhaled small particles are trapped in the ASL and are physically removed by sneezing, nasal blowing, and mucociliary clearance. Mucociliary clearance is a coordinated defense mechanism of the submucosal gland and goblet cells that produce mucus, which ciliated epithelial cells then transport to the nasopharynx. Both CRSwNP and CRSsNP often feature impaired mucociliary clearance [44,45]. Patients with primary ciliary dyskinesia, a genetic disorder characterized by impaired ciliary movement, frequently present with CRS as a concurrent condition [46]. A similar situation is observed in transgenic mice engineered with a deletion of Dnaic1, which mimics primary ciliary dyskinesia and results in CRS [47]. Both inherent factors like genetic mutations in patients with polycystic kidney disease and external pathogens such as pollutants and microbes can cause ciliary dysfunction. Additionally, chronic and repeated episodes of infection and inflammation associated with CRS can lead to a loss of cilia and increased mucus viscosity, thereby resulting in dysfunctional mucociliary clearance [17]. Improvements in impaired mucociliary clearance in CRS, measured using the saccharine test, have been observed after endoscopic sinus surgery [48]. In summary, a decrease in mucociliary clearance is a significant contributor to CRS pathophysiology.

### 4.2. Physical Barriers and Epithelial Cell Dysfunction

SECs provide a protective physical barrier against chemical and physical environmental stimuli. The integrity of the epithelium, which is essential for this protective function, is maintained by cell junctions, including tight junctions, adherens junctions, gap junctions, desmosomes, and hemidesmosomes [49,50]. Apical-junctional complexes form between adjacent cells and consist of the most apical tight junction and the underlying adherens junction. Tight junctions are mainly composed of the apical belt and regulate the paracellular transport of ions. Underlying adherens junctions are responsible for the initiation and maintenance of cell–cell adhesion [51]. The adherens junction interacts with the actin cytoskeleton through the transmembrane protein E-cadherin and the intracellular proteins α-catenin and β-catenin [52]. The junctional proteins controlling apical-basal epithelial cell polarity include claudins, occludin, junctional adhesion molecules, and ZO; these proteins maintain the homeostasis of the mucosal epithelium by controlling structural integrity, ion diffusion, solute and microbe permeation, and cell proliferation and migration [53,54].

SEC barrier dysfunction has been suggested as one of the important mechanisms in the pathogenesis of CRS [21]. Various pathogens like viruses and bacteria, allergens, and even fine particulate matter (≤2.5 μm; PM 2.5) can disrupt the function of the SEC barrier and affect the body’s defenses [55,56]. The integrity of the epithelium appears to be compromised as a result of the reduced expression of tight junction proteins like claudin-4, occludin, and ZO-1, as has been observed in both biopsies and nasal epithelial cell cultures from patients with CRSwNP [29,57,58]. Changes in other junctional proteins such as E-cadherin and claudin-1 as well as in desmosomes have been noted in mucosal samples from patients with CRS [59,60,61]. The transepithelial resistance (TER) of human primary SECs from patients with CRSwNP was found to be significantly lower than the TER of SECs from healthy individuals [58]. External environmental pathogens can cause pathological changes to the sinonasal epithelium. *Pseudomonas aeruginosa* can disrupt occludin and claudin-1 [62] and *Staphylococcus aureus* can disrupt ZO-1 [63]. Inflammatory cytokines such as IL-4 and IL-13 (Th2 cytokines) and interferon (IFN)-γ (Th1 cytokine) influence epithelial integrity and decrease the TER, whereas IL-17 does not [58]. Eosinophils also contribute to barrier dysfunction, as they release granules and form eosinophil extracellular traps when activated [64].

Hypoxic conditions, which could be caused by CRS, lead to a downregulation of ZO-1 and E-cadherin, as well as a decrease in the TER of normal human nasal epithelial (HNE) cells [65], implying that such hypoxic conditions may result in increased epithelial cell permeability. A recent study revealed that early-stage middle turbinate polyps in patients with CRSwNP exhibit more severe epithelial cell loss compared to those with mature ethmoidal polyps and normal nasal mucosa. Furthermore, the reduced expression of epithelial cell junctional proteins, such as E-cadherin, ZO-1, and occludin, in patients with mature ethmoidal CRSwNP indicates a compromised epithelial cell barrier in the polyps, making them more susceptible to damage [23].

## 5. Innate Immune Responses of Epithelial Cells in CRS

### 5.1. Chemical Barriers: Mucus and Defense Molecules

The secretory cells of the submucosal gland of the nasal cavity are composed of serous cells and mucous cells. The water-rich secretions produced by serous cells consist of proteins and peptides such as lysozyme, lactoferrin, and sIgA, which play various antimicrobial roles [66]. On the other hand, mucous cells secrete mucin, a highly glycosylated protein. CRSwNP and CRSsNP are both accompanied by mucin overproduction.

Properly hydrated mucus is important for airway homeostasis. Inadequate mucus hydration, mucus hypersecretion, and alterations in the properties of mucus result in chronic airway inflammation and airway obstruction in patients with cystic fibrosis (CF), chronic obstructive pulmonary disease (COPD), and asthma. Mucins are a group of highly glycosylated macromolecular proteins that synthesize most of the mucus layer and are either secreted or membrane associated [67]. Mucins are evolutionarily differentiated, resulting from more than 20 MUC genes; in general, the distribution of expression varies according to tissue type [68]. MUC5AC and MUC5B are macromolecular gel-forming mucins and are the main components of airway mucus responsible for its viscosity [69]. Under normal conditions, MUC5AC is localized to epithelial goblet cells at low levels, while MUC5B is secreted mainly by mucus cells of the submucosal gland [70,71,72]. MUC5AC has been used as a marker of goblet cell metaplasia [73] and is upregulated in various chronic respiratory inflammatory disorders, such as COPD, CF, and asthma [74,75]. Their electrophoretic behaviors differ [76]. Depending on the ratio of these mucins, mucus with different functional properties can be observed.

High mucus viscosity is correlated with the severity of chronic respiratory inflammatory airway diseases like COPD and CF [77]. It is believed that MUC5AC and MUC5B expression is increased in patients with CRS. However, there have been conflicting reports on whether the expression of MUC5AC and MUC5B is increased or decreased in patients with CRS [70]. These conflicting results may reflect the different endotypes involved. One randomized controlled trial reported that MUC5AC expression was downregulated in patients with both CRSwNP and aspirin-resistant asthma after 2 weeks of oral prednisone therapy followed by 12 weeks of intranasal corticosteroids but not in patients with both CRSwNP and aspirin-intolerant asthma [78].

MUC5B, but not MUC5AC, appears to play a critical role in host defenses against airway inflammatory disorders. It has been reported that MUC5B-knockout mice accumulate mucus in both the upper and lower airways. Phagocytosis was impaired due to the accumulation of apoptotic macrophages, and the production of IL-23 was decreased in these MUC5B-knockout mice compared with MUC5B transgenic mice [79]. In a murine model of COPD, MUC5B promotes goblet cell differentiation by the induced expression of STAT6 and SPDEF and appears to regulate inflammation via macrophage-related mechanisms [80].

Under pathological conditions, inflammatory mediators may induce changes in mucin production. Type 2 and non-type 2 cytokines, growth factors, and lipid mediators can induce the overexpression of MUC5AC [81,82,83,84]. Th2 cytokines, such as IL-4, -5, and -13; MUC5AC; and MUC5B are all significantly overexpressed in patients with IL-5(+) CRSwNP compared to patients with IL-5(−) CRSwNP. Treatment with IL-4 and -13 stimulates MUC5AC and MUC5B secretion in patients with CRSwNP via IL-4α, which is also highly expressed in patients with IL-5(+) CRSwNP [84].

### 5.2. Pattern Recognition Receptors (PRRs)

PRRs are key components of the innate immune system, which is the body’s first line of defense against infection. PRRs are expressed by cells of the immune system including airway epithelial cells (AECs). PRRs can recognize structures common to many pathogens, known as pathogen-associated molecular patterns (PAMPs), as well as damage-associated molecular patterns (DAMPs), which are endogenous signals produced in response to cell damage or stress [85]. Upon recognizing these patterns, PRRs can initiate an innate immune response. Toll-like receptors (TLRs) are membrane proteins that can recognize a broad range of PAMPs. For example, TLR4 recognizes lipopolysaccharide (LPS), a component of the outer membrane of Gram-negative bacteria, while TLR3 recognizes double-stranded RNA, a form of viral genetic material. When PRRs recognize a PAMP or a DAMP, they initiate signal transduction pathways that lead to the activation of transcription factors like nuclear factor-kappa B and interferon regulatory factor (IRF). This process results in the production of various inflammatory and antimicrobial peptides such as cytokines and type I IFNs, which help control the infection and serve to alert and activate the adaptive immune system [86].

Dysregulation of PRR signaling can lead to an inappropriate or overactive immune response, contributing to chronic inflammation and tissue damage. Several studies have reported that the expression of TLRs may be upregulated in the inflamed sinonasal mucosa of individuals with CRS compared to healthy controls [87,88]. Nasal polyps exhibit a reduced expression of antimicrobial PLUNC (palate, lung, nasal epithelium clone) proteins, including SPLUNC-1 [89]. SPLUNC-1 also influences the volume of ASL by blocking the activation of sodium channels in epithelial cells, which are responsible for mediating sodium and fluid absorption in airway epithelia. Pendrin/SLC26A4 is an apically expressed epithelial anion exchanger whose expression is reportedly upregulated in patients with CRSwNP [90]. Pendrin/SLC26A4 also influences ASL volume and could potentially have a significant impact on mucociliary clearance.

A major component of Gram-positive bacteria may contribute to the upregulation of bradykinin receptor 1 in nasal mucosa, which plays a role in the progression of CRSsNP. A robust, positive correlation has been observed between the Gammaproteobacteria members *Haemophilus*, *Enterobacter*, and *Pseudomonas* and the activity of TLR2/1 and TLR4 [91].

## 6. Epithelial Cell–ECM Crosstalk

Understanding the communication between epithelial cells and the ECM has emerged as a crucial factor in the pathophysiology of CRS. Epithelial cells, which serve as a protective barrier, also engage with the ECM, contributing to tissue homeostasis and responses to pathologic stimuli.

In CRS, a disruption of this delicate balance can result in aberrant tissue remodeling characterized by an abnormal deposition of the ECM. As previously discussed in Section 3, TGF-β acts as a pivotal mediator in this interaction. On one hand, TGF-β regulates the synthesis of ECM components such as collagen, fibronectin, and proteoglycans. It also modulates the activity of MMPs and their tissue inhibitors, which are essential players in ECM degradation and remodeling [92]. TGF-β also influences the differentiation of T-cells into regulatory T-cells, adding to the diversity of the patterns of inflammation observed in patients with CRS [20].

Beyond its role in tissue remodeling and regulatory T-cell differentiation, TGF-β is also important in driving Th17 cell differentiation. TGF-β, in conjunction with pro-inflammatory cytokines like IL-6 or IL-21, activates STAT3 and promotes the differentiation of Th17 cells [93]. Within the CRS landscape, augmented Th17 responses steered by TGF-β can intensify inflammation, marked by an upsurge in MMPs, which leads to the breakdown of basement membranes and the ECM [94]. The subsequent surge in IL-17 production from Th17 cells can magnify the inflammatory response in the ECM, detrimentally affecting epithelial cell functionality and integrity [95]. This highlights the dynamic and multi-dimensional influence of TGF-β, which not only orchestrates tissue remodeling but also significantly shapes the inflammatory environment in CRS.

Moreover, the crosstalk between epithelial cells and the ECM extends to immune responses. Innate immunity might affect the histological features and associated cytokines in CRS-related tissue remodeling via TLR2 and TLR4 [96]—effects that are also evident in the differential regulation of TLRs in response to inflammatory stimuli. TLR2 expression shows a negative correlation with squamous hyperplasia in patients with CRSsNP, but a positive correlation with gland hyperplasia is observed in patients with CRSwNP. Interestingly, while TGF-β1 expression is decreased by a TLR2 agonist in patients with CRSwNP, it is increased by a TLR4 agonist in patients with CRSsNP; MMP-9 is also upregulated in such cases [96].

Recent evidence suggests that the impairment of fibrinolysis, which leads to excessive fibrin deposition, plays a crucial role in the formation of nasal polyps [97]. Activation of the plasminogen activator/plasmin system is also important for tissue remodeling during the development of nasal polyps. Protease-activated receptors (PARs) are expressed in AECs, including those found in the nasal epithelium Thrombin, and PAR-1 agonists contribute to the formation of nasal polyps by stimulating the production of the vascular endothelial growth factor (VEGF) in AECs [98]. TGF-β regulates thrombosis by acting as an inducer of plasminogen activator inhibitor 1 (PAI-1) [99]. At both the transcriptional and protein levels, TGF-β1, PAI-1, and tissue-type plasminogen activators are found to be decreased in both early-stage and mature nasal polyps compared to healthy sinonasal epithelium [100]. In nasal mucosa-derived fibroblasts of patients with CRSsNP, thromboxane A2 plays a key role in controlling the expression of chemokines CXCL1 and CXCL8, which are found at high levels in the nasal mucosa of patients with CRSsNP. Meanwhile, the thromboxane A2 receptor is expressed abundantly, not only in CRSsNP mucosa but also in control mucosa [101].

This complex interplay between epithelial cells and the ECM underscores the need to understand the molecular mechanisms governing this interaction. Future studies focused on unraveling these mechanisms could provide novel therapeutic targets for the management of CRS.

## 7. Nasal and Airway Epithelial Cell–Immune Cell Crosstalk

The delicate balance underlying the health of our nasal and airway passages is maintained by a dynamic interplay between epithelial cells and various immune cells. These interactions provide the first line of defense against pathogens, ensuring prompt and effective immune responses. Epithelial cells act as barriers, sentinels, and communicators, coordinating with immune cells to mediate inflammation, initiate repair, and fine-tune immune reactions. This section delves into the multifaceted crosstalk between nasal and airway epithelial cells and key immune cells, including macrophages, neutrophils, DCs, B-cells, T-cells, and eosinophils. By understanding these interactions, we gain insights into the pathogenesis of conditions like CRS and potential therapeutic strategies.

### 7.1. Nasal and Airway Epithelial Cell–Macrophage Crosstalk

Macrophages, predominant in the airway mucosa, exhibit complex interactions with epithelial cells for tissue remodeling and fibrosis. Depending on environmental cues and specific stimulants, monocyte and tissue macrophages can transition into either pro-inflammatory (M1) or anti-inflammatory (M2) roles. Factors such as TLR ligands, IFN-γ, and GM-CSF primarily direct macrophages towards the M1 phenotype, characterized by the release of pro-inflammatory mediators like TNF-α, IL-6, and IL-1β [102]. These M1 macrophages are known for promoting the Th1 immune response. The released inflammatory agents can stimulate fibroblasts, which further elevate the production of agents like IL-6 through mechanisms such as the NK-kB pathway [103,104]. This heightened activity by the fibroblasts supports tissue fibrosis, either by their transmutation into myofibroblasts or by increasing their cell numbers.

Conversely, cytokines IL-4 and IL-13 drive the differentiation of macrophages into the M2 phenotype [102]. In CRSwNP, there is a notable increase in M2 macrophages, which are readily activated by TGF-β1. Their increased presence has been shown to be correlated with elevated levels of type 2 mediators like IL-5, total IgE, and ECP [105]. Moreover, M2 macrophages release a cocktail of growth factors like TGF-β1, VEGF, and PDGF, which can amplify the destruction of the ECM and tissue remodeling [106]. The TIM-4 molecule also plays an important role in determining macrophage behavior in CRSwNP, specifically through TGF-β1-mediated EMT [107]. In ECRSwNP, there is a pronounced inclination towards M2 macrophages collaborating with eosinophils. In stark contrast, non-ECRSwNP appears predominantly influenced by M1 macrophages and activated CD4+ memory T cells [108]. Interestingly, during the early stages of nasal polyps in CRS, there is a notable surge in subepithelial eosinophils and M2 type macrophages when contrasted with mature polyps and healthy mucosa [57]. The delicate interplay between M1 and M2 macrophages and their epithelial interactions shapes the progression of CRS. Grasping this balance is essential for honing targeted treatments for sinus-related disorders.

### 7.2. Nasal and Airway Epithelial Cell–Neutrophil Crosstalk

AECs are responsible for secreting CXCL8 (IL-8), a powerful chemokine that attracts neutrophils. When the airway epithelium is damaged due to allergen exposure, CXCL8 is released and binds to CXCR1 and CXCR2 receptors on neutrophils [109]. Also involved in neutrophil recruitment, IL-17A is a cytokine that promotes inflammation to help the body defend itself against pathogens. When IL-17A is overproduced, it can lead to inflammation and tissue damage [110]. IL-17 is produced by Th17 lymphocytes, which in turn upregulate the epithelial cell expression of CXCR2 chemokines. This process leads to the recruitment of leukocytes to the airway [111].

### 7.3. Nasal and Airway Epithelial Cell–DC Crosstalk

The epithelium and DCs work together to defend against inhaled pathogens. DCs are found either on or beneath the basal lamina of the epithelium. Research has demonstrated that communication between epithelial and DCs affects cell function during inflammation [112]. The epithelium produces thymic stromal lymphoprotein (TSLP), IL-33, GM-CSF, and CCL-20, which can bind DCs and activate T-cells when antigens are present [113,114]. Notably, both TSLP and IL-33 function as DAMPs, molecules released by stressed or damaged cells, signaling the innate immune system to initiate a response. Their release can be triggered by tissue damage, infection, or inflammation, emphasizing their pivotal role in modulating the immune response [115]. The expression of TSLP, IL-33, and their receptors are increased in ECRS rather than non-ECRS [116]. DCs process antigens and present them to T- and B-lymphocytes, helping the immune system respond to the external environment. A study was conducted using a triple-cell co-culture model consisting of nasal epithelial cells, DCs, and macrophages. It was found that the production of TSLP was significantly higher in the co-cultured nasal epithelial cells of patients with asthma compared to control subjects [117]. TSLP stimulates DCs to increase the expression of CD40, OX40, and CD80, leading to Th2 inflammation [118]. In nasal polyps, there is a significant increase in the number of plasmacytoid (pDCs) and myeloid (mDCs) DCs compared to those in noninflamed nasal mucosa. However, in more severe cases (nasal polyps with asthma), the number of pDCs decreases, whereas the number of mDCs remains the same [119]. To develop better treatments for nasal polyposis, it is important to study how subsets of DCs behave in different environments and how they affect T-cell differentiation.

### 7.4. Nasal and Airway Epithelial Cell–B-Cell Crosstalk

Nasal and airway epithelial cells possess the capability to internalize antigens from their immediate environment. Once these antigens are absorbed, they can be conveyed directly to underlying B cells or relayed through DCs. Such interactions facilitate the activation of B-cells, leading to heightened antibody production [120]. The mucosal tissue of CRSwNP exhibits an increased presence of naïve memory B cells, plasma cells, and antibodies, compared to healthy controls [121,122]. This B cell infiltration is distinctly more pronounced in CRSwNP than in CRSsNP, as observed with other immune cells, such as cytotoxic CD8+ T cells and macrophages [123]. While the tissue levels of antibodies like IgM, IgG, IgA, and IgE are more abundant in CRSwNP [120], a heightened IgD level characterizes CRSsNP [124].

In CRS, especially CRSwNP, autoreactive B cells produce antibodies that target nuclear antigens, such as double-stranded DNA (dsDNA), and basement membrane components [125,126]. These elevated anti-ds-DNA IgG and IgA levels are linked with severe and recurrent CRSwNP manifestations [127,128]. Notably, nasal polyp tissues show pronounced complement activation, with C4d deposition particularly evident at the basement membrane [129]. Whether this C4d localization arises from autoantibodies targeting the basement membrane or specific pathogens remains a topic of ongoing research.

The airway epithelium can impact B-cell responses through the presence of a B-cell-activating factor (BAFF) and proliferation-inducing ligand (APRIL) [130]. BAFF and APRIL promote the maturation, survival, and proliferation of B-cells, leading to the production of IgG, IgA, and IgE. Patients with asthma show an increase in both BAFF expression and B-cell number in their sputum and serum, indicating that there may be a connection between BAFF and airway inflammation [131]. When mucosal epithelial cells, macrophages, and eosinophils produce BAFF and APRIL, B-cell activation is triggered, leading to IgE secretion, which in turn causes a Th2 immune response [132,133]. BAFF is found in both the nasal epithelium and subepithelial layer [134]. Epithelial cells help transport immunoglobulins produced by B-cells to the mucosal surface [130]. The levels of BAFF expression in children with allergic rhinitis were notably higher than levels in normal children, suggesting the potential use of BAFF in developing new treatment methods [135]. In polyps, there is a significant increase in the number of BAFF+ cells, CD20+ cells (B-cells), and CD138+ cells (plasma cells) [134]. In patients with CRSwNP, serum BAFF levels are increased, which correlates with the severity of eosinophil infiltration in the mucosa [136]. Furthermore, the serum BAFF level was identified as a useful biomarker for differentiating the subtypes of CRSwNP before surgery, as well as for predicting the likelihood of recurrence after surgery [136].

### 7.5. Nasal and Airway Epithelial Cell–T-Cell Crosstalk

The airway epithelium produces four co-stimulatory T-cell molecules of the B7 family. These molecules help either initiate or prevent specific interactions between T-cells and other cells by acting as ligands [137]. B7-H1, also known as programmed death-ligand 1 (PD-L1 or CD274), and B7-DC, known as programmed death-ligand 2 (PD-L2 or CD273); both bind to programmed death-1 (PD-1). They generate a signal that inhibits T-cells [138]. Memory T-cells are activated by B7-H2, whereas CD4+/CD8+ cells are activated by B7-H3. HNE cells cultured at the air–liquid interface showed high transcript levels of B7-H3 and B7-H2, but HNE and B7-H2 activities were not significantly changed during differentiation [139]. Stimulation with IFN-γ induced the expression of B7-H1 and B7-DC in both BEAS2B and primary HNE cells [140]. Immunohistochemical staining confirmed the presence of B7-H1, B7-H2, and B7-H3 in epithelial cells, particularly in patients with Samter’s Triad, a severe form of CRS [140].

Beyond co-stimulatory molecules, nasal and airway epithelial cells produce an array of chemokines and cytokines that can recruit and regulate T cell subsets [141,142]. In the backdrop of allergic inflammation, certain cytokines secreted by the epithelium, including TSLP, IL-25, and IL-33, skew the T-cell responses predominantly towards a Th2 phenotype. IL-4 emerges as a central mediator here, promoting the Th2 lineage from naïve Th0 lymphocytes and concurrently inhibiting apoptosis in activated T cells. A deeper dive into the pathways underlying these effects highlights the importance of the IL-4-induced IRS-2/PI-3 K/PKB and Jak/STAT cascades, particularly in terms of their anti-apoptotic effects on CD8+ T cells [143]. Further consolidating the Th2 response are chemokines CCL1, CCL17, and CCL22, whose interactions with specific T cell receptors accentuate allergic responses [144]. Th2 cytokines, such as IL-4 and IL-13, upregulate CCL17 expression in nasal polyp epithelium, whereas Th1 cytokines increase CCL17 levels in both normal and nasal polyp epithelium [145].

However, epithelial cells are not restricted to Th2 modulation. Pathogens often stimulate a Th1 response. Epithelial-derived interferons, particularly IFN-γ, catalyze Th1 responses, aiding in the differentiation and recruitment of Th1 cells [146]. Epithelial cells can also release chemokines like CXCL16, which not only help in the recruitment of Th1 cells but also modulate group 2 innate lymphoid cell (ILC) migration, thereby influencing pathways like ERK1/2, p38, and Akt [147]. The resultant Th1 milieu is rich in pro-inflammatory cytokines like IFN-γ and IL-2, which play pivotal roles in intracellular pathogen clearance [148].

Moreover, epithelial cells, through the secretion of IL-6, can drive naive T cell differentiation towards the Th17 phenotype. The signature cytokine of Th17 cells, IL-17, has gained attention in the CRS narrative, particularly in the context of neutrophilic inflammation. Even though the intricacies of Th1/Th17 responses in relation to epithelial modulation are not as defined as Th2, disruptions in the epithelial integrity or inflammatory provocations appear to manifest an enriched Th1/Th17 cytokine milieu in CRS [95].

Regulatory T (Treg) cells, known for their immune-modulating properties, are key in maintaining the delicate balance between Th1/Th2 and Treg/Th17 [149]. While Tregs themselves do not show a consistent elevation or reduction in CRSwNP versus CRSsNP [22,150,151], their functional impairment is evident. Current data suggest a discrepancy in the number of Tregs between the peripheral blood and the sinonasal tissue in CRSwNP, potentially due to the impaired migration of Tregs toward nasal epithelial cells [152]. More specifically, Tregs in CRSwNP patients demonstrate a reduced chemotactic response to the chemokines CCL1 and CCL17 produced by nasal epithelial cells, which might explain their altered distributions in tissue [145,153]. Importantly, the balance between Tregs and Th17 cells, another subset of T cells with pro-inflammatory characteristics, is crucial in CRS pathology [149]. These two cell types, which share some common differentiation pathways, exhibit a complex interplay: while TGF-β can promote the differentiation of both [154], their transcriptional regulators, RORγt/RORα and FOXP3, counteract and suppress the expression of one another [155,156]. Notably, the dysregulation of this Treg-Th17 balance, influenced by various factors, including microbial flora imbalances, plays a significant role in the pathogenesis and progression of CRSwNP [157].

### 7.6. Nasal and Airway Epithelial Cell–Eosinophil Crosstalk

In the airway, high numbers of eosinophils can lead to inflammation and damage. Chronic inflammation can be caused by interactions between AECs and eosinophils in these diseases [158]. Eosinophils play a role in the development of asthma, COPD, and rhinosinusitis. They have direct effects on AECs and can lead to increased mucus production and airway remodeling [159]. Eosinophils have higher levels of oxidases than neutrophils, and they can damage the airway epithelium via direct cytotoxicity during the respiratory burst response [160]. The chemokines eotaxin 1 (CCL11), eotaxin 2 (CCL24), eotaxin 3 (CCL26), and RANTES (CCL5), which are primarily released from the epithelium, regulate eosinophil recruitment to the airway. The release of CCL24 and CCL26 by AECs is stimulated by epithelial differentiation in response to Th2 cytokines [161]. Elevated levels of IL-4 and CCL26 have been found in ECRSwNP [162]. The stimulation of IL-4 has been found to lead to an increase in the stability of CCL26 mRNA, and the expression of CCL26 has been observed to be affected by the overexpression or knockdown of the human antigen R (HuR) [162], an RNA-binding protein of the ELAV protein family. HuR plays a crucial role in various biological processes such as immune responses, cell proliferation, and differentiation [162]. Patients with type 2 asthma associated with eosinophilia express 15-lipoxygenase 1 (15LO1) in AECs. The increased expression of 15LO1 was observed in the epithelial cells of nasal polyps; 15LO1 also correlates with CCL26 expression and colocalizes with CCL26 in basal cells. Thus, 15LO1 could be considered a novel therapeutic target for CRSwNP treatments [163].

### 7.7. Nasal and Airway Epithelial Cell–Mast Cell Crosstalk

Mast cells, intricately linked with the pathology of CRS, are known for their capacity to orchestrate the recruitment of eosinophils and basophils and to release mediators that influence vasodilation and tissue edema. A particular focus has been on tryptase-positive mast cells in the context of ECRSwNP. Both Takabayashi et al. and Cao et al. have identified elevated numbers of these cells in the epithelial regions of affected patients, emphasizing their potential role in disease progression [164,165].

The precise dynamics between the epithelium and mast cells in CRS remains enigmatic. However, prevailing theories suggest that epithelial cells might generate a spectrum of cytokines upon exposure to certain stimuli. This could act as a prompt, nudging mast cells and related immune entities to discharge Th2-centric cytokines [4]. Adding another layer to this intricate narrative is the observed proliferation of basophils, traditionally viewed in tandem with mast cells, in the matrix of nasal polyp tissues [166]. The presence and interaction of these cells with the epithelial domain may possibly play a decisive role in dictating sinus disease trajectories. Delving deeper into this interaction could unveil new therapeutic strategies for CRS management.

### 7.8. Nasal and Airway Epithelial Cell–ILC Crosstalk

ILCs have recently garnered attention for their pivotal role in orchestrating immune responses, particularly in mucosal barriers like the nasal and airway epithelium. Unlike adaptive immune cells, ILCs do not rely on antigen receptors for activation. Instead, they respond swiftly to cytokines and other environmental cues, making them essential first responders. ILCs, especially ILC2s, are highly attuned to epithelial cell-derived signals. Upon stress or injury, epithelial cells release proinflammatory cytokines, such as IL-25, IL-33, and TSLP. While these were initially recognized as the primary activators of ILC2s, it is now evident that ILC2s possess an array of receptors on their surface. The engagement of these receptors with their ligands can also stimulate ILC2s to produce type 2 cytokines [167]. This underlines the versatility and adaptability of ILC2s in their crosstalk with the epithelial environment. Furthermore, transcription factors, notably nuclear factor kappa-Β (NF-κB), NFAT, STAT5, and STAT6, are instrumental in modulating type 2 cytokine production by ILC2s [168]. Their activation offers another layer of regulation, ensuring that ILC2 responses are tailored to the specific needs of the epithelial tissue.

While both ILC2s and Th2 cells are involved in type 2 immunity, their interactions with epithelial cells and modes of activation differ significantly. Th2 cells require antigen recognition through T-cell receptors and antigen presentation by DCs, which introduces a delay in their response. Conversely, ILC2s, with their array of receptors and rapid response to epithelial proinflammatory cytokines, offer a swift and early inflammatory reaction, acting as the immune system’s first responders.

## 8. Therapeutic Approaches Impacting the Sinonasal Epithelium in CRS

CRS presents a multifaceted therapeutic challenge due to its varied etiological factors. Topical therapies, primarily topical corticosteroids, play a key role by reducing epithelial inflammation, by promoting repair, and by preventing further barrier dysfunction. Saline irrigations, another topical approach, cleanse the epithelial layer, promoting mucus clearance and symptom relief.

Corticosteroids, through their binding to glucocorticoid receptors (GCRs) in sinonasal epithelial cells, wield potent anti-inflammatory actions, especially through the influence of the active isoform GCRα on inflammation-related gene transcription [169]. The efficacy of corticosteroids, however, can be impacted by the expression of the varying isoforms of GCR. Intranasal corticosteroids, often delivered via nasal irrigation, are the primary post-surgery treatments for CRS, being proven to significantly alleviate inflammatory symptoms [169]. Concurrently, a short course of oral steroids, as an adjunct to intranasal variants, has shown efficacy in enhancing the sinus microenvironment and shrinking nasal polyps [170]. The innovative exhalation delivery system with fluticasone (EDS-FLU) optimizes corticosteroid delivery, facilitating significant improvement in various CRS symptoms and overall patient quality of life [171].

For CRSwNP, which often remains uncontrolled despite standard therapies, biological interventions targeting pro-inflammatory mechanisms have emerged as viable treatments [172]. Mepolizumab, an anti-IL-5 agent, showed marked improvement in nasal polyp size and symptomatology when administered over a year [173]. Similarly, tezepelumab, targeting the cytokine TSLP, exhibited efficacy in reducing eosinophil counts and asthma exacerbations when used as an adjunct therapy for CRS [174]. Dupilumab, inhibiting IL-4 and IL-13 signaling, effectively improved various patient scores and inflammation markers, even outperforming mometasone nasal spray in reducing polyp burden [175]. Additionally, omalizumab, an anti-IgE biologic, enhanced symptom management in CRSwNP patients with limited intranasal steroid responses, and its benefits were observed over extended periods [176]. As the understanding of CRS endotypes deepens, these biologics present opportunities for precision treatment in the future.

For patients unresponsive to medical therapies, surgery remains an option. Endoscopic sinus surgery provides dual benefits, primarily clearing obstructed sinuses and optimizing the epithelial surface for enhanced topical drug delivery. It’s crucial, however, to perceive it as part of a holistic management strategy. Looking ahead, emerging therapies targeting the restoration of the epithelial barrier offer hope. Additionally, innovative strategies seeking to modulate the sinonasal microbiome and gaining further understanding of intricate epithelial cellular interactions may open new therapeutic frontiers. As research progresses, a shift towards personalized medicine that allows for tailored treatments based on specific epithelial responses or associated biomarkers (thereby ensuring a precise and effective therapeutic approach) appears inevitable.

## 9. Conclusions

Overall, the role of epithelial cells in CRS goes beyond forming a simple physical barrier. Epithelial cells function as an active immune system component in nasal and sinus cavities. Their immunological role is multifaceted and crucial in the initiation and regulation of immune responses. Epithelial cells play important roles in the prevention, development, and progression of CRS by actively communicating with immune cells and orchestrating immune responses (Figure 3). An understanding of these immunological processes can aid in the development of better therapeutic strategies for CRS management.

## Figures and Tables

**Figure 1 ijms-24-14229-f001:**
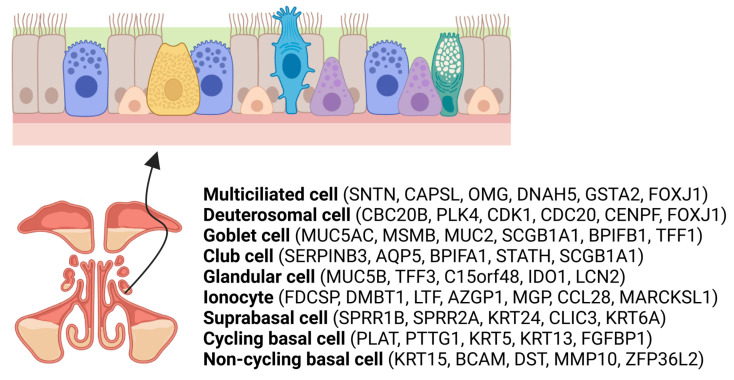
Overview of the nasal epithelium, showing various cell types (black arrow). The epithelial cells consist of a variety of cell types, including multiciliated, deuterosomal, goblet, glandular, ionocyte, suprabasal, cycling basal, and non-cycling basal cells. These cells undergo changes in composition and transcriptomics depending on the presence of disease.

**Figure 2 ijms-24-14229-f002:**
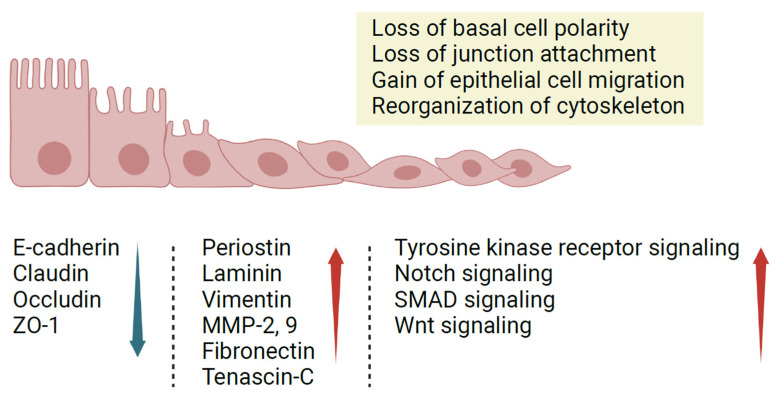
Epithelial-Mesenchymal Transition (EMT) in the Nasal Epithelium. When an injury or inflammation occurs, basal cells lose their epithelial characteristics, such as cell–cell junctions and apical-basal polarity. Further, epithelial cells lose their characteristics and develop mesenchymal traits. They start expressing mesenchymal markers (red arrow), and there is a significant decrease in the expression of epithelial markers like E-cadherin (blue arrow). This can contribute to wound healing or the progression of a disease, depending on the individual’s health status.

**Figure 3 ijms-24-14229-f003:**
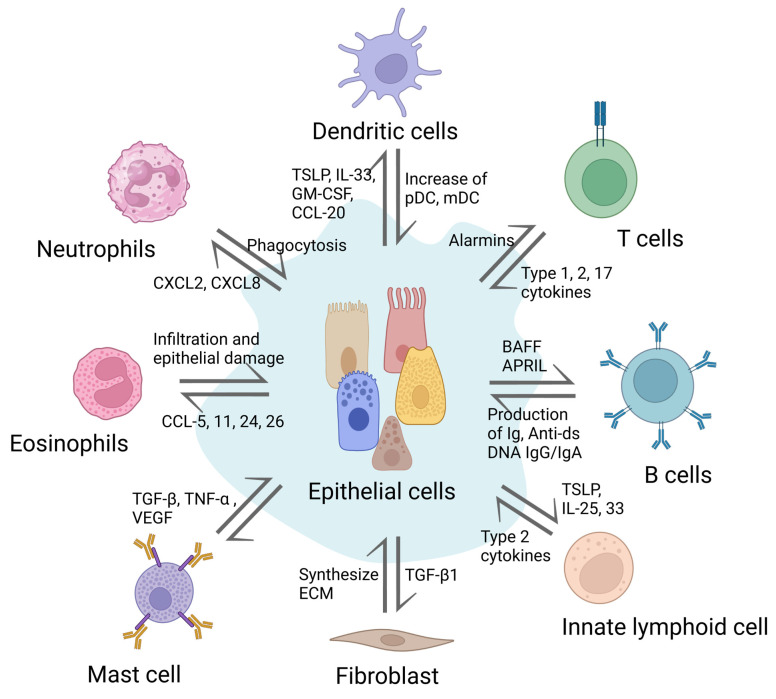
Interaction between nasal epithelial cells and various immune cells during an immune response in the nasal cavity. During an immune response in the nasal cavity, nasal epithelial cells interact with different immune cells, resulting in the production of various cytokines and chemokines. These substances act as mediators for the immune response and help to attract more immune cells to the area. This can either increase or decrease inflammation.

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
