# Peer review of "Unraveling the Role of Epithelial Cells in the Development of Chronic Rhinosinusitis"

_ijms, 2023, doi:10.3390/ijms241814229_

Round 1
Author Response
(REVIEWER 1)
This is a well-written and well-structured article, reviewing the role of epithelial cells in the pathogenesis of chronic rhinosinusitis comprehensively.
Here are some questions and suggestions for the manuscript.
Would the heading of “2. Histological features of the sinonasal epithelium” be more specific if it were changed to “the cell composition of the sinonasal epithelium,” “overview of sinonasal epithelial cell subtypes,” or other more appropriate heading?
- Thank you for pointing this out. As you recommended to change the heading of the chapter, we changed it as “overview of sinonasal epithelial cell subtypes.”
Can you address the EMT process in CRSsNP? Are there differences in the EMT process between CRSsNP and CRSwNP?
- Thank you for highlighting the significance of the EMT process in CRSsNP. EMT, or epithelial-to-mesenchymal transition, plays a crucial role in the remodeling of sinonasal tissues, especially in the context of chronic rhinosinusitis (CRS). In CRSsNP (CRS without nasal polyps), the EMT process is believed to be present, but less dominant, compared to CRSwNP (CRS with nasal polyps). EMT markers, such as decreased E-cadherin expression and increased vimentin, are less pronounced in CRSsNP tissues, indicating a subdued EMT process.
- Contrastingly, in CRSwNP, an enhanced EMT process, potentially driven by the increased presence of growth factors like TGF-β, has been observed. This could be a contributing factor to the polyp formation that characterizes CRSwNP.
- To address your query further, while both CRS subtypes involve the EMT process, its intensity and the resultant tissue remodeling vary, with CRSwNP showing a more pronounced EMT-driven pathology. We have updated our manuscript to incorporate this information and provide a clearer distinction between the EMT processes in CRSsNP and CRSwNP.
On page 6, section 5.1., the abbreviation “sIgA” should be written as “secretory IgA”.
- Thank you for pointing this out. We have made the suggested correction and have now used the term "secretory IgA" instead of the abbreviation "sIgA" in section 5.1 on page 6.
Reference #39 does not seem relevant to the content.
On page 7, section 5.2., the transcription factor of the IFN signaling pathway is interferon
regulatory factor (IRF).
- It was our mistake. We modified it as you pointed out.
Many citations in section 7 are based on bronchial epithelial cell studies, would it be better to change the heading to “airway epithelial-immune cell crosstalk”? In addition, do the authors believe that the results of experiments on bronchial epithelial cell can be representative of nasal epithelial cells? Why or why not?
- We appreciate the suggestion to consider changing the heading to “airway epithelial-immune cell crosstalk.” The broader term "airway" certainly encompasses both nasal and bronchial regions. However, our primary focus in this chapter was on nasal epithelial interactions. While we did utilize some bronchial epithelial cell studies to draw parallels and provide comprehensive insights, the core emphasis remains on nasal epithelium. If deemed more accurate, we could consider revising the title to "Nasal and Airway Epithelial-Immune Cell Crosstalk" to encapsulate the scope of both nasal and bronchial studies.
- We believe that while bronchial and nasal epithelial cells have distinct anatomical locations and functions, they also share many structural and immunological characteristics due to their primary role as barriers in the respiratory system. Indeed, several signaling pathways, receptors, and immune responses are conserved between these cell types. As a result, findings from bronchial epithelial cell studies can offer valuable insights into nasal epithelial cell behavior, especially when direct nasal epithelial data might be limited. That said, we also acknowledge that there are unique aspects to each cell type, and direct extrapolation can sometimes be overly simplistic. Regarding nasal epithelial function, we sought to utilize bronchial studies as supportive and complementary rather than definitive.
Macrophages also play an important role in the chronic airway inflammation, is there evidence for crosstalk between airway epithelial cells and macrophages?
- Thank you for your comment. We discussed crosstalk between airway epithelial cells and macrophages in section 7.1 Considering that macrophages are immune cells that encounter inhaled pathogens most quickly, this discussion is placed at the forefront.
In Figure 3, the authors listed references to interactions between epithelial cells and individual types of immune cells and fibroblasts. However, the figure would be more informative if the key mediators produced by epithelial cells to attract or influence immune cells and the mediators secreted by immune cells that act on epithelial cells were addressed.
- In light of your suggestion, we have made an effort to incorporate more detailed information into the figure.

Reviewer 2 Report
In their review article, "Unraveling the Role of Epithelial Cells in the Development of Chronic Rhinosinusitis", the authors describe the multifaceted and complex role that epithelial cells have in different forms of CRS. The article is clearly written, easy to understand and the English language is excellent. The topic itself is vast and not easy to cover in a single review article - it is understandable that compromises must be made in the depth of introducing each subtopic. Some sections seem to be more comprehensive than others. Especially the immunology section suffers, as it would be challenging even if the entire article focused on different immune cell types alone.
Minor comments:
Section 5-1: mucins are introduced twice as highly glycosylated proteins, remove the repetition. Some text rearrangement would improve the section: MUC5A and B are introduced in the first paragraph but the text would fit more properly the next paragraph that reintroduces them again.
Section 5-2: "When PRRs recognize a PAMP or a DAMP, they initiate signal transduction pathways that lead to the activation of transcription factors like nuclear factor-kappa B and IFN". Should IFN rather be IRF (Interferon Regulatory Factor), which is a transcription factor rather than a cytokine? It is unclear, what reference is used here, number 57?
Section 6: TGF-beta is an immunoregulatory cytokine and important for Treg development, but it is also a critical Th17 cytokine. This and some of it's implications in CRS could be mentioned in the text.
Section 7-1: I would replace the word "chemical" (when referring to CXCL8) by "chemokine", for accuracy.
Section 7-2: The DC section discusses TSLP and IL-33, but does not bring forward their function as DAMPs.
Section 7-4: "The B7-H1 and B7-DC proteins bind programmed death-ligand 1 (PD-L1) and PD-L2, respectively." Please check this sentence. B7-H1 is PD-L1 and B7-DC is PD-L2 and they both bind PD-1. These proteins are most often called with the PD-name or Cluster of Differentiation number (CD274 etc.), so I recommend using either and mentioning aliases.
Major comments:
Figure 1 and the text related to it: Several rare cell types are mentioned here. many of these are likely unknown to the reader. Some brief explanation of at least some of them would have been interesting. The picture itself is not very informative with just the cell names without any explanation of a function.
Section 7: As mentioned above, it is not an easy task to include most immune cell types to a review without lengthening the text too much. However, broadening the topic here would benefit the article.
The introductory paragraph doesn't serve very well as an introduction to epithelial cell - immune cell crosstalk topic: it goes to transcriptional level details on random cell types, and yet is very short. It would be better if it was more summarizing and general, and the details would be placed elsewhere.
Some of the sections that discuss different immune cells seem to be focused quite much on a narrow field of research: most notably, B cells and BAFF/APRIL, and T cells and B7 immune checkpoint molecules. I would add broader discussion of different functions. Some extra detail of, for example the abovementioned molecules, can be swapped for this if needed.
I would gladly see a stronger emphasis of how the function of the different cells is known to differ in the known types of CRS - especially Th2 vs. Th1/17. The last picture in the review shows only reference numbers, which are already easily found in the clearly titled paragraphs above. A picture that shows an actual summary of the roles of the cells (maybe even in different types of CRS) would be a lot more interesting. This kind of a picture could also show some cross-talk of the immune cells in the context of communication with epithelia cells, which would show the reader more clearly that the immunological factors underlying CRS are indeed a complex network.
Mast cells and ILCs would deserve both an own paragraph - the ILCs especially due to their rather recently discovered, paramount role in immunity.
Lastly, the therapeutic possibilities for CRS are quite thinly discussed. The authors could add a short section for this important part of the topic.
Author Response
(REVIEWER 2)
In their review article, "Unraveling the Role of Epithelial Cells in the Development of Chronic Rhinosinusitis", the authors describe the multifaceted and complex role that epithelial cells have in different forms of CRS. The article is clearly written, easy to understand and the English language is excellent. The topic itself is vast and not easy to cover in a single review article - it is understandable that compromises must be made in the depth of introducing each subtopic. Some sections seem to be more comprehensive than others. Especially the immunology section suffers, as it would be challenging even if the entire article focused on different immune cell types alone.
Minor comments:
Section 5-1: mucins are introduced twice as highly glycosylated proteins, remove the repetition. Some text rearrangement would improve the section: MUC5A and B are introduced in the first paragraph but the text would fit more properly the next paragraph that reintroduces them again.
- We removed the repetition as you recommended. We also checked and modified reference numbers in section 5-1, which were inappropriately matched (ref 67-84).
Section 5-2: "When PRRs recognize a PAMP or a DAMP, they initiate signal transduction pathways that lead to the activation of transcription factors like nuclear factor-kappa B and IFN". Should IFN rather be IRF (Interferon Regulatory Factor), which is a transcription factor rather than a cytokine? It is unclear, what reference is used here, number 57?
- The reference to IFN was indeed incorrect in the context of transcription factors initiated by PRRs upon recognizing PAMPs or DAMPs. I have since modified the text to correctly reference "Interferon Regulatory Factor" (IRF) instead of IFN, and I also listed new reference number 86 (Li, D., Wu, M. Pattern recognition receptors in health and diseases. Sig Transduct Target Ther 6, 291 (2021). https://doi.org/10.1038/s41392-021-00687-0). I appreciate your keen observation, and such feedback is invaluable in ensuring the accuracy and clarity of our work.
Section 6: TGF-beta is an immunoregulatory cytokine and important for Treg development, but it is also a critical Th17 cytokine. This and some of its implications in CRS could be mentioned in the text.
- We appreciate for your kind comments. Discussion of the impact of TGF-beta to Th17 cytokine was additionally described in section 6.
Section 7-1: I would replace the word "chemical" (when referring to CXCL8) by "chemokine", for accuracy.
- We replaced the word “chemical” by “chemokine.”
Section 7-2: The DC section discusses TSLP and IL-33, but does not bring forward their function as DAMPs.
- We additionally described the role of TSLP and IL-33 as DAMPs.
Section 7-4: "The B7-H1 and B7-DC proteins bind programmed death-ligand 1 (PD-L1) and PD-L2, respectively." Please check this sentence. B7-H1 is PD-L1 and B7-DC is PD-L2 and they both bind PD-1. These proteins are most often called with the PD-name or Cluster of Differentiation number (CD274 etc.), so I recommend using either and mentioning aliases.
- We appreciate your diligence in highlighting this oversight. We have revised the section to read:
- "The B7-H1, also known as programmed death-ligand 1 (PD-L1 or CD274), and B7-DC, known as programmed death-ligand 2 (PD-L2 or CD273), both bind to programmed death-1 (PD-1)."
- Your feedback is invaluable, and we have taken note of your recommendation to use the PD or CD designations for clarity and consistency.
Major comments:
Figure 1 and the text related to it: Several rare cell types are mentioned here. many of these are likely unknown to the reader. Some brief explanation of at least some of them would have been interesting. The picture itself is not very informative with just the cell names without any explanation of a function.
- Thank you for your insightful feedback on Figure 1 and the associated text. We've added concise explanations for each of the rare cell types in the main text, elucidating their primary function, markers, and significance in the sinonasal epithelium context. This should provide clarity and a better foundation for understanding subsequent discussions in the article.
Section 7: As mentioned above, it is not an easy task to include most immune cell types to a review without lengthening the text too much. However, broadening the topic here would benefit the article.
The introductory paragraph doesn't serve very well as an introduction to epithelial cell - immune cell crosstalk topic: it goes to transcriptional level details on random cell types, and yet is very short. It would be better if it was more summarizing and general, and the details would be placed elsewhere.
Some of the sections that discuss different immune cells seem to be focused quite much on a narrow field of research: most notably, B cells and BAFF/APRIL, and T cells and B7 immune checkpoint molecules. I would add broader discussion of different functions. Some extra detail of, for example the abovementioned molecules, can be swapped for this if needed.
- We added broader discussion of the crosstalk between airway epithelium and B cells/T cells.
I would gladly see a stronger emphasis of how the function of the different cells is known to differ in the known types of CRS - especially Th2 vs. Th1/17. The last picture in the review shows only reference numbers, which are already easily found in the clearly titled paragraphs above. A picture that shows an actual summary of the roles of the cells (maybe even in different types of CRS) would be a lot more interesting. This kind of a picture could also show some cross-talk of the immune cells in the context of communication with epithelia cells, which would show the reader more clearly that the immunological factors underlying CRS are indeed a complex network.
- Thank you for taking the time to provide such detailed and insightful feedback. We appreciate your emphasis on the distinction between the functions of different cells in the various types of CRS, especially concerning Th2 and Th1/17.
- We've taken your comments to heart and have made the necessary modifications to the review. The figure has been revised to provide a clearer summary of the roles of the cells. We've also incorporated the cross-talk of the immune cells with epithelial cells to better reflect the complex immunological network underlying CRS. We believe these changes align well with your suggestions and will enhance the clarity and depth of our review.
Mast cells and ILCs would deserve both an own paragraph - the ILCs especially due to their rather recently discovered, paramount role in immunity.
- We added chapter 7-1 Nasal and airway epithelial cell-macrophage crosstalk.
Lastly, the therapeutic possibilities for CRS are quite thinly discussed. The authors could add a short section for this important part of the topic.
- Thank you for pointing out the importance of therapeutic possibilities for CRS. We have now included an additional discussion on therapeutic approaches that impact the sinonasal epithelium. Given the breadth and complexity of this topic, our aim was to present an overview and introduce the general concepts within the scope of this chapter.

Round 2
Reviewer 2 Report
Thank you for the well-written revision for the review article. It's quality has now significantly increased and I have no further requests for modifications. Unfortunately in the pdf, only Figure 2 was visible to me. However, if the authors have modified the two other pictures as they say they have, I believe they are suitable for the publication.